# Suberin Fatty Acid Hydrolysates from Outer Birch Bark for Hydrophobic Coating on Aspen Wood Surface

**DOI:** 10.3390/polym14040832

**Published:** 2022-02-21

**Authors:** Anuj Kumar, Risto Korpinen, Veikko Möttönen, Erkki Verkasalo

**Affiliations:** 1Natural Resources Institute Finland, Production Systems, Tietotie 2, 02150 Espoo, Finland; risto.korpinen@luke.fi; 2Natural Resources Institute Finland, Production Systems, Yliopistokatu 6B, 80100 Joensuu, Finland; veikko.mottonen@luke.fi (V.M.); erkki.verkasalo@luke.fi (E.V.)

**Keywords:** birch bark, wood coating, suberin fatty acids, hydrophobicity, artificial aging

## Abstract

Bark extracts are sustainable sources of biopolymers and have great potential to replace fossil-based polymers in wood coating applications. The present study investigated the applicability of suberin fatty acids hydrolysate (SFA) extracted from the outer bark of silver birch (*Betula pendula* Roth.) for coating of aspen wood (*Populus tremula* L.). The SFA combined with maleic anhydride (MA) and octadecyltrichlorosilane (OTS) as a curing agent was prepared in ethanol and used in surface coating. The water contact angle, surface reflectance spectra, FTIR, and SEM-EDS were used to characterize the physical and chemical properties of the coated wood samples. Further, the long-term stability of the SFA coating was analyzed via artificial aging. The wood surface became hydrophobic, as the contact angle for the water droplet (WCA) was over ~120°, and was stable for all of the prepared combinations of SFA, MA, and OTS.

## 1. Introduction

As a hydrophilic, hygroscopic, porous, and fibrous material, wood is especially vulnerable to water sorption, because water penetrates rapidly into the wood structure causing swelling and eventually a loss of mechanical strength as well as providing conditions for biological degradation [1,2]. Several wood modification methods have been developed to improve the overall chemical and physical properties of the material. Wood modification can be divided into two main categories: active and passive wood modification. Active modification includes chemical, thermal, and enzymatic modifications of the wood cell wall and surface, whereas passive modification consists mainly of impregnation of the pores and lumen space of the cell wall with chemicals [3,4]. Wood surface coating is one of the active chemical modification techniques to improve the service life of wood and wood products, and various synthetic polymeric coatings have been investigated for this purpose [2,5]. However, in recent years, polymers and monomers derived from bio-based materials have attracted enormous interest due to the dwindling of non-renewable feedstock such as petrochemicals [6,7].

The wood processing industry provides large amounts of bark as a side stream, and most of it is incinerated for energy instead of cascading for potential green chemicals [8,9,10]). The global annual generation of bark is estimated to be between 300 and 400 million m^3^ [11]. In birch bark, in particular, the chemical composition of the outer and inner bark differs considerably. The outer bark of silver birch (*Betula pendula* Roth.) is made up of 45% suberin, 40% extractives, 9% lignin, 4% hemicelluloses, and 2% cellulose, on average [12]. The extractives mainly consist of different triterpenoids but only few phenolics—in contrast to inner bark with its very high phenolic contents [13,14]. Birch wood is especially rich in the valuable biochemicals such as the naturally occurring biopolyester suberin and the triterpenoids betulinol and lupeol [12,15]. The content of betulinol and lupeol in outer bark varies between 30% and 35%, and the content of suberin can be up to 40–50% [16]. Birch bark containing both inner and outer bark from a pulp mill has been reported to contain 5.9% suberin [16]. Suberin fatty acids have various functional applications such as a hybrid co-polymer and polyols for polyurethane [17,18], a source of organosolv lignin [19], a thermoset resin from epoxy ω-hydroxyacids and methacrylates [20], a water vapor barrier hybrid film of cellulose [21], and as a hydrophobic coating for a lignocellulosic fiber network [15]. Handiso et al. [22] evaluated the physico-chemical properties of cellulose surface modified with suberin and suberin fatty acids, and the outcomes from this strongly support the use of SFA in cellulose modification for better hydrophobic and barrier properties.

In this study, the suberin fatty acids hydrolysate (SFA) of outer birch bark was investigated as a potential hydrophobic coating application on aspen wood surface. Two crosslinking agents, maleic anhydride (MA) and octadecyltrichlorosilane (OTS), were used in several combinations. The coated wood samples were evaluated by Fourier transform infrared spectroscopy (FTIR), surface reflectance spectra, and scanning electron microscopy with energy-dispersive X-ray analysis (SEM-EDX). In addition, the water contact angle was measured using the sessile drop method. Further, the effectiveness of the hydrophobic properties of wood samples was evaluated by exposing them to artificial aging.

## 2. Materials and Methods

### 2.1. Materials

The outer bark of silver birch (*Betula pendula* Roth.) was manually removed from freshly cut stems (tree diameter: 200–300 mm), followed by air-drying at room temperature. The outer bark was then ground using a cutting mill Fritsch PULVERISETTE Type 15.903 (Fritsch GmbH, Germany) with a sieve cassette having 4 × 4 mm^2^ square openings. The ground outer bark was then freeze-dried and stored in an airtight polyethylene bag.

European aspen wood (*Populus tremula* L.) was procured from a local sawmill in southern Finland and prepared as specimens free from knots and cracks with straight grains and normal growth ring widths. The wood samples were prepared to uniform dimensions (100 × 45 × 5 mm) and oven-dried at 105 °C for 24 h prior to coating.

Isopropyl alcohol 99.8% (Merck KGaA, Darmstadt, Germany), sodium hydroxide 99.0% (Merck KGaA, Darmstadt, Germany), ethanol 94.0% (Altia Oyj, Rajamäki, Finland), and sulfuric acid 95% (VWR International S.A.S., Briare, France) were used in the experiments. Then, 2 molar sulfuric acid was prepared by pouring 112 mL of concentrated sulfuric acid into 600 mL of maxima ultrapure water. The final volume was adjusted with ultra-pure water to 1000 mL after the solution was cooled to room temperature.

### 2.2. Suberin Fatty Acids Extraction Process

The dry solid content of the ground and dried outer bark was 97.8%. The hydrolysis and extraction were carried out according to a protocol described by Korpinen et al. (2019) [15]. In the process, 150 g o.d. of outer bark was used with a liquid-to-bark ratio of 10 mL/g. Ethanol was used as the solvent with a solvent-to-water ratio of 9:1 (*v*/*v*). On o.d. bark, 20% NaOH was applied, and the bark was boiled in the solution for 3 h. The betulinol fraction and suberin fatty acids were then separated. The extractions were repeated three times. The different fractions were characterized separately and mixed before the wood surface treatment.

### 2.3. Wood Samples and Coating Solutions Preparation

The wood samples without any coating were selected as a control (**C**). Figure 1 shows a schematic of the extraction and coating process of SFA on wood surface. The following SFA-based coating formulations were prepared and applied on the wood surface:SFA and maleic anhydride (MA) (treatment name, **SFA–MA**): Fixed amounts (70:30) of SFA and maleic anhydride as a hardener were mixed and boiled at 45 °C in ethanol. A manual brush coating was used to prepare a uniform two-layer coating of SFA–MA on wood and dried for 24 h at 105 °C;Treatment of coated samples with octadecyltrichlorosilane (OTS) (treatment name, **SFA–MA + OTS**): SFA–MA-coated samples dipped into a solution (1/100) (*v*/*v*) of OTS and n-hexane for 30 min and dried for 24 h at 105 °C;**OTS** treatment of sample: Wood samples dipped in a solution (1/100) (*v*/*v*) of OTS and n-hexane for 30 min and dried at 24 h at 105 °C;**SFA–OTS**: In 100 mL ethanol, 5 g SFA and 500 µL OTS were mixed and boiled at 120 °C (10 min) and cooled down to room temperature and applied manually to the wood surface (two coatings).

### 2.4. GC and GC-MS Analysis

Suberin fatty acids were derivatized using a mixture of N,O-bis(trimethylsilyl)trifluoroacetamide (BSTFA):chlorotrimethylsilane (TMCS):pyridine (120:20:20) and heated for 50 min at 70 °C. The silylated samples were quantified by gas chromatography flame ionization detection (GC-FID), and the peak identities were confirmed by gas chromatography mass spectrometry (GC-MS) as described by Korpinen et al. (2019) [15].

### 2.5. SEM and EDS Analysis

The surfaces of coated wood samples and uncoated reference samples were evaluated using scanning electron microscopy (SEM, JEOL JSM-5500LV, Osaka, Japan). The specimens were coated with a thin evaporated layer of carbon to improve conductivity prior to the analysis. The prepared specimens were mounted 12 mm from lenses. The chamber pressure < 1.0 × 10^−5^ (high vacuum) and acceleration voltages of 15 and 10 kV were used to measure the specimen’s surface. Further, energy-dispersive X-ray spectroscopy (EDS) was applied with ESPRIT v.2.1.2.17832 coupled with SEM.

### 2.6. Color Measurement

The surface reflectance spectra of coated wood samples and uncoated reference samples were measured at intervals of 8 mm in diameter for each specimen in the visible light wavelength range of 360–740 nm using a Konica Minolta CM-2600d spectrophotometer (Konica Minolta Inc., Tokyo, Japan). Spectral data were converted to CIEL*a*b* color coordinates using a 2° standard observer and a D65 light source for lightness (L*), redness (a*), and yellowness (b*), according to the CIEL*a*b* color space (ISO 11664-4:2008). For each sample group, the mean and standard deviations of the color coordinates were calculated.

One-way ANOVA was performed using the Origin (Lab 9.1, MA, USA) statistical data analysis software to calculate the samples’ mean difference at *p* < 0.05 levels for the color differences in samples before and after artificial aging.

### 2.7. Water Contact Angle Measurement

The wettability of the coated wood samples and uncoated reference samples was evaluated based on the water contact angle (WCA) using the sessile drop method. The contact angles were measured by means of computer-aided analysis (OCA 50 instrument coupled with SCA 20 CA software, DataPhysics Instruments, Berlin, Germany) of elliptical shapes of liquid drops as observed in an optical goniometer and recorded by a digital camera installed in axial extension of the lens and statistics of 5 droplets per specimen. Drops of approximately 4–6 μL, the volumes being calculated from images of the drops, were applied by means of a dispenser.

### 2.8. ATR-FTIR Measurement

Fourier transform infrared spectrophotometer (Shimadzu Cooperation, Kyoto, Japan, IRPrestige-21/IRAffinity-1/FTIR-8000 series) coupled with IRsolution software were used to gain, control, and process the data. Semi-thin layers were cut from wood specimens with a razor blade and then dried at 60 °C for two hours. The prepared specimens were scanned while using an attenuated total reflection (ATR) setup in the absorbance range of 400–4000 cm^−1^ with a scanning rate of 2 cm^−1^ and 50 scans per run.

### 2.9. Artificial Aging

The artificial aging of coated wood samples and uncoated reference samples were performed to compare and understand the long-term performance of prepared coating formulations. The aged samples were characterized to see the changes due to the fact of aging. The artificial aging was carried out according to ISO 4892-2:2013 using an Atlas Xenotest 440 weather testing chamber (Ametek Inc., Baton Rouge, LA, USA) for 500 h. Xenon light exposure at 60 Wm^−2^ in the range of 300–400 nm at 65 °C was used (black standard).

## 3. Results and Discussion

### 3.1. Analysis of the SFA Hydrolysates

Two integral and characteristic peaks of a long aliphatic chain in suberin appeared at 2919 and 2851 cm^−1^ as shown in Figure 2. A high-intensity band vibration at 1738 cm^−1^ of the carbonyl groups of typical esters was present in suberin. The symmetric and asymmetric bonds associated with vinyl groups, such as C–O and C–H, were present at 1245, 1164, and 722 cm^−1^. The characteristic epoxy band was present at 845 cm^−1^ of hydroxyl-fatty acids, i.e., 9,10-epoxy-18-hydroxy-18:0 acid [23,24], which accounted for ~10% of the dry birch outer bark [25]. The bands of saturated aliphatic chains of vinyl (C–H) in-plane bending were present at 1463 cm^−1^ [22].

The chemical composition of the suberin fatty acids in the hydrolysate can be seen in Table 1. The monomer composition was similar to the results obtained by Pinto et al. (2009) [10]. The main difference was found in the amount of 9,10,18-trihydroxy-18:0 acid; the trihydroxy acid was formed during the hydrolysis in ethanol instead of isopropanol. The chlorinated substances were most probably artefacts from the silylation of the samples for the chemical analysis. The small amounts of triterpenoids, such as betulinol and betulinic acid, were derived from the first filtration process, where the betulinol fraction was separated [9]. The filtration process was not fully complete, but further separation techniques are required if fully pure fractions are needed.

### 3.2. Analysis of SFA Hydrolysates-Based Coatings

#### 3.2.1. ATR-FTIR

The infrared spectra of uncoated reference aspen wood (Sample C) showed absorption bands at 3345 cm^−1^ of the O–H group stretching vibration from the intermolecular and intramolecular hydrogen bonds [26] in Figure 3. The C–H stretching absorption between 2850 and 2930 cm^−1^ in the methyl and methylene groups [1,27] and the fingerprint region between 1800 and 900 cm^−1^ showed the absorption bands of functional groups present such as C–O at 1732 and C–C at 1505 cm^−1^ [28]. Several other important vibration modes present in lignin and carbohydrates were also observed at 1157, 1107, and 1052 cm^−1^, etc. [28]. The OTS-coated sample showed characteristics peaks at ~2850 and ~2920 cm^−1^ for terminal methyl groups with different intensities and Si–O–C at 1192 cm^−1^. The OTS treatment usually deposited the hydrophobic self-assembled monolayers via a hydrolysis and condensation process [1]. In a wood polymer system, the OTS is hydrolyzed, most likely by free –OH groups, while layered n-alkylsiloxane (PODS) gels are formed and polymerized into OTS layers on the wood surfaces [29].

In this study, two crosslinking agents were used to crosslink the hydroxy groups present in the SFA hydroxylates. At first, MA was used as a crosslinking agent, and the SFA underwent a maleation reaction due the reaction with MA and maleated SFA having two-way interactions with hydroxy groups; it reacted with epoxides first and the remaining hydroxy groups later [15]. After SFA-based coating with different combinations of crosslinkers, the FTIR spectra showed the formation and changes in the characteristics of the functional groups of the wood surface as shown in Figure 3. The stretching peak at 1030 cm^−1^ associated to cellulose C6–O6H C–O stretching in lignin shifted to a lower absorbance intensity, and a major shift appeared in the SFA–OTS combination. Similar to this, several other functional groups (i.e., 1165 cm^−1^ (C1–O–C4′ antisymmetric stretching in cellulose and hemicellulose); 1462 cm^−1^ (lignin C–H bending); 1736 cm^−1^ (C=O stretching); aliphatic chain (CH3 and CH3 at 2919 and 2851 cm^−1^) were more visible in the SFA–OTS coating system. The peak at 1248 cm^−1^ associated with C–O stretching in carboxylic acid shifted to a lower absorbance intensity after all the different coatings, possibly due to the bonding with reactive moieties of SFA and crosslinking agents. Another important characteristic peak belonged to SFA’s ester group at 1736 cm^−1^, and hemicellulose (C=O) significantly shifted to a higher peak absorbance for the SFA–OTS coating system, while it did not change for other combinations of coatings. Importantly, the OH stretching at ~3350 cm^−1^ also changed after SFA-based coating systems.

#### 3.2.2. SEM-EDS

The SEM images of the longitudinal section of the samples are presented in Figure 4. The uncoated reference sample C showed a defibrillated longitudinal surface, because we used rough wood samples with no surface finishing to obtain a better penetration of the coating and clearly visible open voids. The EDS scanning mainly showed the presence of the chemical compounds of carbon (62.24 atom%) and oxygen (37.76 atom%). The SEM micrograph of sample SFA–MA looked similar to the reference sample C; however, EDS scanning revealed a change in the chemical compositions of the C atom% that increased from 62.24% to 65.75% and of the O atom% that decreased from 37.76% to 34.35%. The changes were mainly due to the interaction of SFA–MA and free hydroxyl groups on the wood surface [15].

The OTS coating seemed to reduce the surface roughness of specimens due to the deposition of self-assembly monolayers [1,29]. EDS scanning confirmed the deposition of silicon (0.35 atom%), residual chlorine (0.30 atom%) as well as increased carbon (62.45 atom%) and reduced oxygen (36.90 atom%). It was also confirmed by the ATR-FTIR results via the formation of S–O–C bonding at 1192 cm^−1^ that the OTS hydrolyzed first and reacted with free –OH groups (see also [1]). The SEM-EDS of the sample SFA–MA + OTS provided a different prospective on the coated wood surface, as the wood was first coated with SFA–MA and the second layer of OTS was formed after curing on the surface. The SEM image showed a uniform surface, and most of the roughness and voids were filled due to the coatings. EDS scanning confirmed significant changes in the atom% of carbon and oxygen, and silicon and chlorine also appeared due the additional OTS coating. Similar changes were observed in SEM-EDS scanning for the sample coated with SFA–OTS composition.

#### 3.2.3. Water Contact Angle (WCA)

The mean water contact angle of the uncoated reference (sample C) was 60 ± 10°, and it was reduced to 17° after 7 s from when the water droplet was absorbed onto the wood (Figure 5). The WCA of the OTS sample was 140 ± 5°, and it remained stable even after 60 s (139 ± 5°). This change occurred due to the deposition of hydrophobic layers of an alkyl group and silicon groups [1]. The SFA–MA coating also increased the wood’s hydrophobicity due to the deposition of long-chain fatty acids. The WCAs of the coated specimens were 126 ± 5°, reducing slightly to 124 ± 2° after 60 s. A similar behavior was observed when cellulosic paper was coated with an SFA–MA formulation [15,22]. The combination of SFA–MA and OTS coating improved the WCA slightly in comparison with SFA–MA only, as it is clearly shown in Figure 4. The SFA–OTS combination provided the highest WCA value of all the protocols tested, being 142 ± 5° and remaining constant for 60 s.

#### 3.2.4. Color

In the color measurement, the uncoated reference (sample C) averaged 89 ± 2 for lightness, 2 ± 1 for redness, and 21 ± 2 for yellowness (Figure 6). After OTS coating, the color of the wood’s surface did not change significantly. Instead, significant changes were observed after SFA–OTS coating, the lightness reducing to 67 ± 3, redness increasing to 7 ± 2, and yellowness increasing to 27 ± 3. The SFA hydrolysates were the main reason behind the color changes. Similar changes were observed in other coating combinations where SFA was a part of the coating. The largest color changes were observed for the SFA–MA-coated samples. Interestingly, the second coating of OTS on the sample SFA–MA + OTS had the largest impact on the color of the wood’s surface. The SFA-based coatings changed the color of the wood surface, making it more reddish and yellowish.

### 3.3. Artificial Aging

#### 3.3.1. ATR-FTIR

Comparison of FTIR spectra of unexposed and aged wood surfaces with different coatings are shown in Figure 7. The significant changes in the lignin aromatic structure were observed due to the exposure to xenon light in the uncoated reference (sample C). The two observed changes appeared at 1596 (aromatic skeletal vibration plus C=O stretch) and at 1510 cm^−1^ (aromatic skeletal vibration). The intensity of all these bands and peaks almost diminished due to the exposure to UV light [30]. The vibration peak at 1268 cm^−1^ of syringyl ring vibration and the C–O stretch of lignin and xylan also disappeared due to the fact of aging under xenon light. On the other hand, the change in the polysaccharides’ absorption bands at 1157 (C–O–C vibration in cellulose and hemicellulose), 1107, and 1052 cm^−1^ in comparison to the unexposed sample [28].

In the case of the coated samples, significant changes after aging were observed in the SFA–MA samples. The peaks associated to C–H stretching absorption between 2850 and 2930 cm^−1^ in the methyl and methylene groups disappeared, and the intensity vibration peak at 1268 cm^−1^ increased significantly, the results being associated with the presence of a syringyl ring. The peak associated to the carbonyl group at approximately 3400 cm^−1^ showed an increased intensity due to the fact of aging for the uncoated refence (sample C), OTS samples, and SFA–MA samples. The samples being coated with combinations, including SFA and OTS, showed high stability in terms of surface chemistry alteration due to the fact of aging.

#### 3.3.2. Water Contact Angle

Usually, most of the protection coating on wood surfaces deteriorates with time due to the continuous change in external environmental conditions; therefore, the water contact angle was measured after the artificial aging of wood surfaces (Figure 8). The WCA of all samples decreased after 500 h of aging. The uncoated reference (sample C) became highly hydrophilic due to the degradation of the aromatic lignin, the WCA being ~47°, and the water droplet being immediately absorbed into the wood. However, all the coating formulations showed a better hydrophobicity with a WCA of over 115° after 500 h of aging. The OTS and its combination with SFA showed a stability towards the aging and all such samples showed a WCA over 120°. Maybe the presence of silico, oxygen, and carbon in the OTS coating provided resistance towards the artificial aging [1].

#### 3.3.3. Color

As can be seen in Figure 9 and Figure 10, the color of the wood surfaces changed much due to the aging. The uncoated reference (sample C) became whiter due to the aging, as yellowness decreased with time. A similar behavior was observed in the OTS samples. In the SFA-based samples, the lightness increased with the aging time; in contrast, especially yellowness but also redness decreased.

## 4. Conclusions

In this study, the suberin fatty acid hydrolysates (SBF) extracted from outer birch bark with a majority composition of 9,10-epoxy-18-hydroxy-18:0 acid was investigated as bio-based polymeric materials for wood coating applications. SFA alone was not able to form any bonding with the wood surface; thus, two types of crosslinking agents, MA and OTS, were used. Different combinations of SFA with OTS and MA were used for wood coating, i.e., OTS alone, SFA–OTS, SFA–MA, and SFA–MA + OTS. The physico-chemical analysis revealed that all SFA-based coating combinations significantly improved the wood surface’s hydrophobicity. The use of OTS as a crosslinking agent for SFA was investigated for the very first time in this study. Even a layer of OTS on the SFA–MA coat contributed to the hydrophobicity of the coating. The color of the wood surface changed notably with the different coatings. Further, the long-term stability of coated wood samples was evaluated using artificial aging up to 500 h in a xenon weather testing chamber, and the aged samples were evaluated for physico-chemical properties. The results showed that all of the coating combinations were stable. The color changes were most notable in the SFA-based coatings: the lightness increased and, especially, the yellowness, but also the redness decreased. Yellowness decreased also in the OTS-coated and uncoated woods, making the wood lighter.

In terms of crosslinker impact on an SFA-coating formulation, MA is mostly used in bio-based polyester crosslinking and is relatively successful in various application. OTS was used for the very first time as a crosslinker for SFA, and it performed better than MA. However, the combination of both MA and OTS performed better in comparison to all combinations. The extraction process of suberin fatty acids from birch bark could affect the quality of bio-based polyesters. Thus, important considerations need to be taken into account during the extraction process in the future course of research work. In conclusion, suberin fatty acid hydrolysates from birch bark could be potential bio-based, renewable substitutes for fossil-based polymeric materials to be used in wood coating.

## Figures and Tables

**Figure 1 polymers-14-00832-f001:**
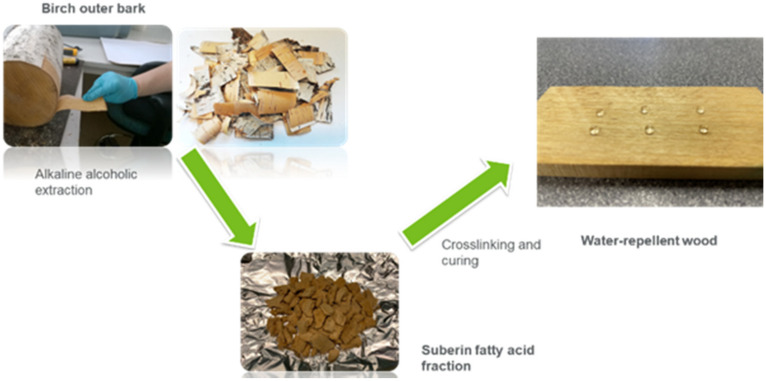
Schematic of SFA extraction and hydrophobic coating on wood.

**Figure 2 polymers-14-00832-f002:**
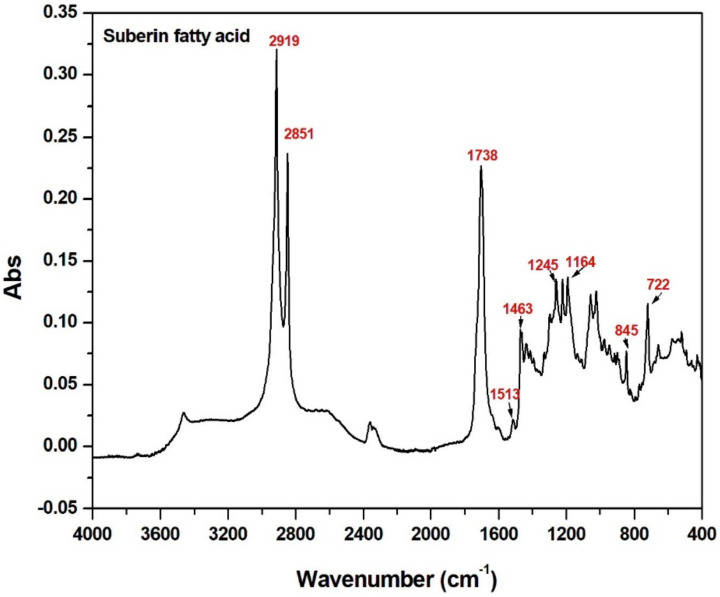
FTIR ATR spectrum corresponding to the SFA hydrolysate.

**Figure 3 polymers-14-00832-f003:**
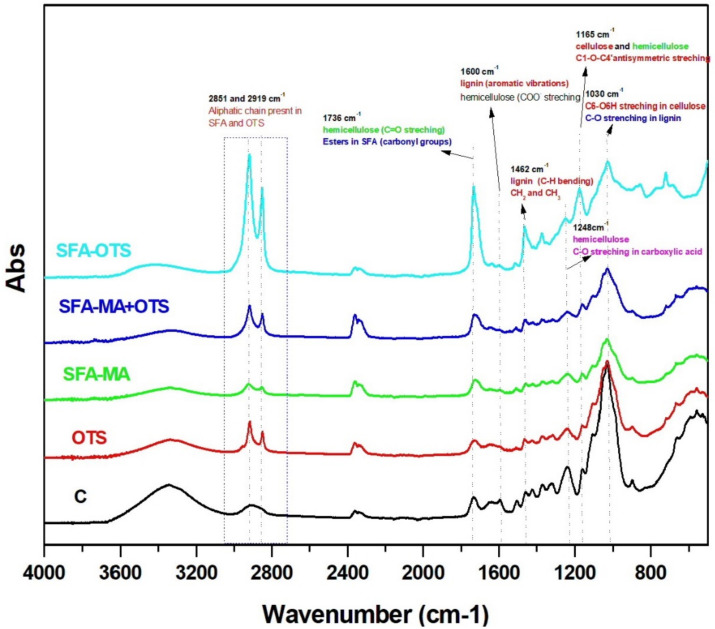
ATR-FTIR spectra of aspen wood coated with SBF hydrolysate-based coatings and uncoated reference wood (sample C).

**Figure 4 polymers-14-00832-f004:**
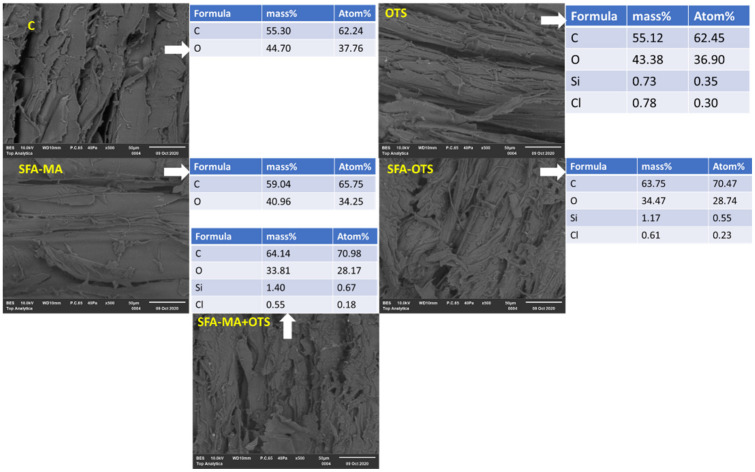
SEM-EDS analysis of aspen wood coated with hydrolysate-based coatings and an uncoated reference wood (sample C). The SEM micrographs were captured at a scale of 100 µm. The EDS scanning was performed at the same scale and surface.

**Figure 5 polymers-14-00832-f005:**
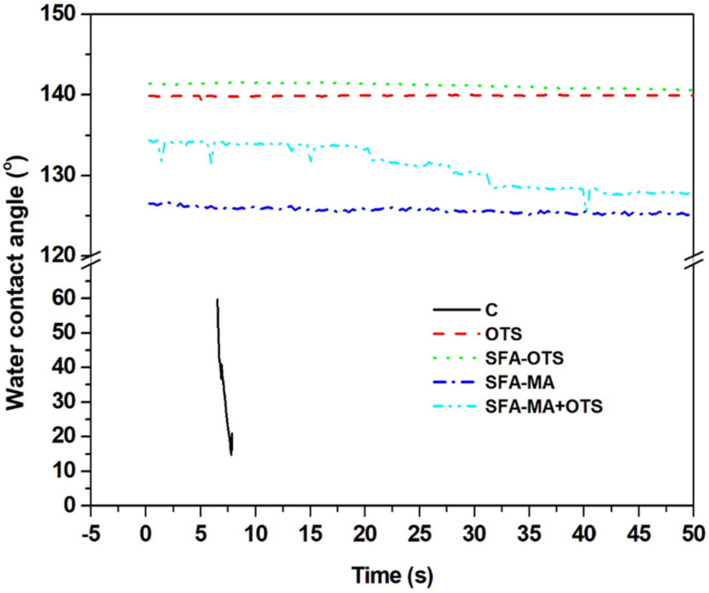
Water contact angle (WCA) of aspen wood coated with SBF hydrolysate-based coatings and an uncoated reference wood (sample C) according to time after applying water droplets on the wood’s surface.

**Figure 6 polymers-14-00832-f006:**
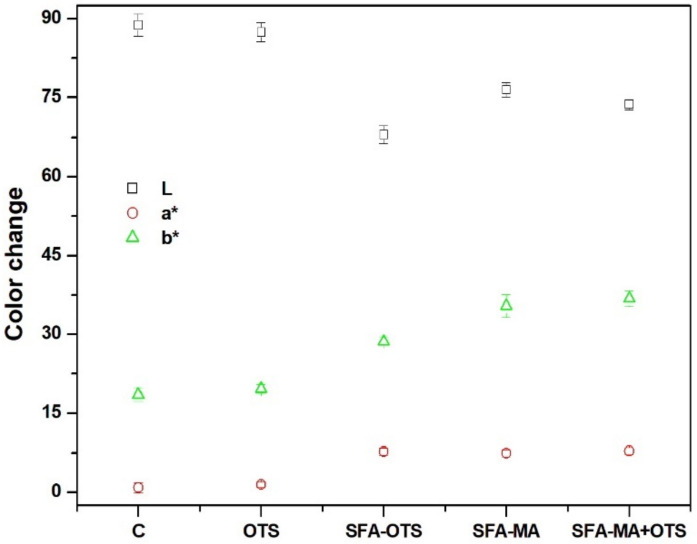
Color of aspen wood coated with SBF hydrolysate-based coatings and an uncoated reference wood (sample C) using the Lab coordinate system, representing lightness (L*), redness (a*), and yellowness (b*).

**Figure 7 polymers-14-00832-f007:**
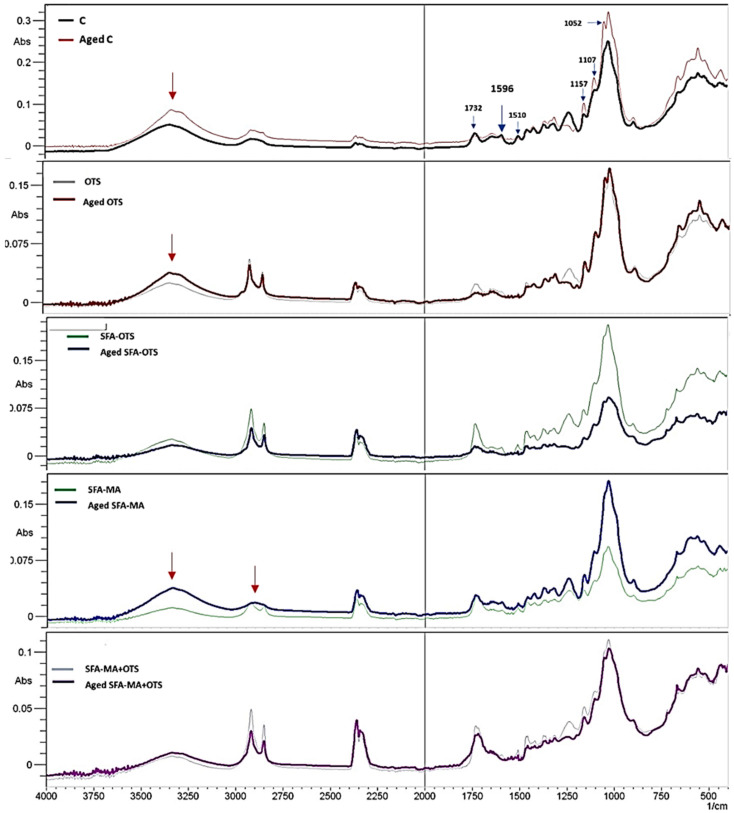
ATR-FTIR spectra of aspen wood coated with SBF hydrolysate-based coatings and artificially aged (500 h) samples.

**Figure 8 polymers-14-00832-f008:**
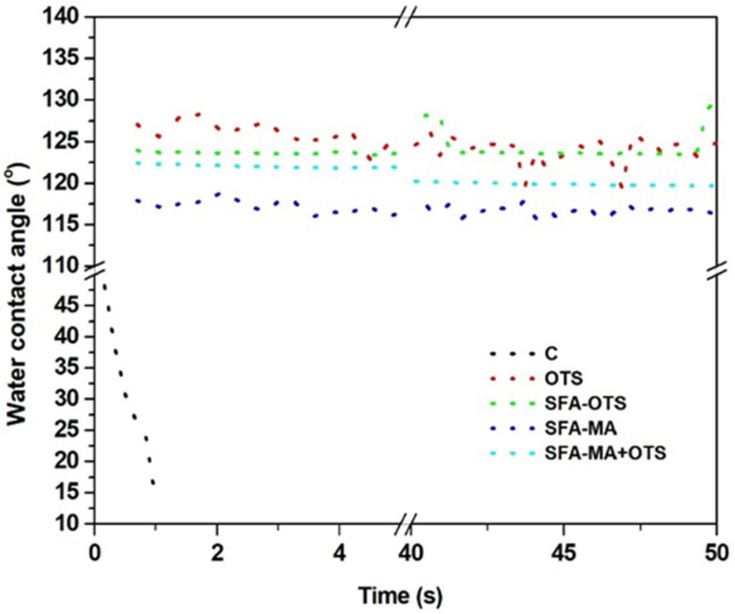
Water contact angle (WCA) of artificially aged (500 h) aspen wood coated with SBF hydrolysate-based coatings and an uncoated reference wood (sample C) according to time after applying water droplets on wood surface.

**Figure 9 polymers-14-00832-f009:**
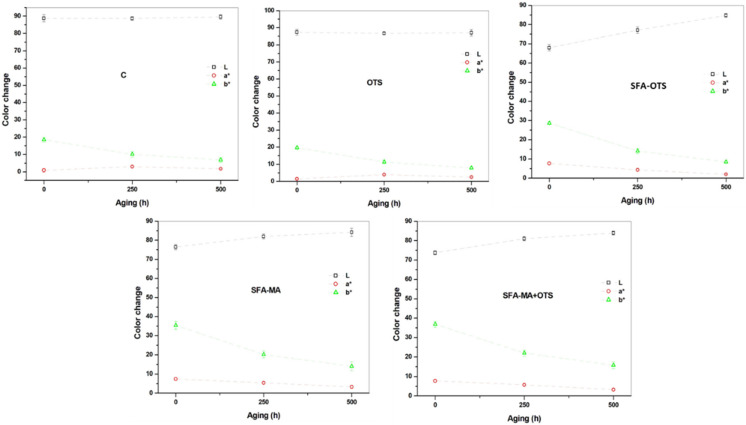
Color of aspen wood coated with SFA hydrolysate-based coatings and an uncoated reference wood (sample C) after artificial aging for 0, 250, and 500 h by xenon exposure using the Lab coordinate system, representing lightness (L*), redness (a*), and yellowness (b*).

**Figure 10 polymers-14-00832-f010:**
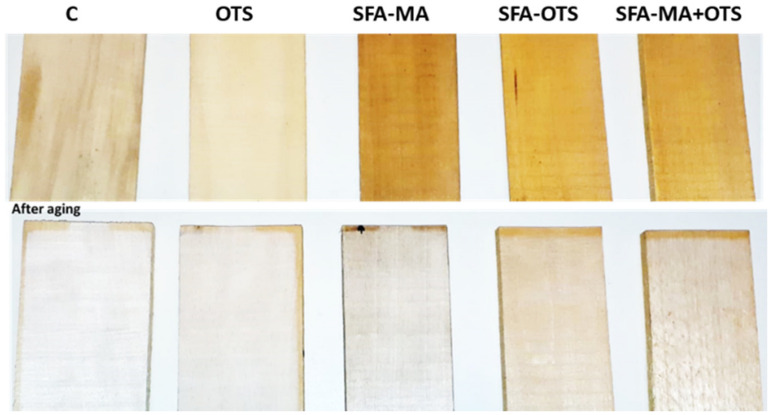
Examples of aspen wood coated with SBF hydrolysates-based coatings and an uncoated reference wood (sample C) before and after artificial aging (500 h) by xenon light exposure.

**Table 1 polymers-14-00832-t001:** The composition of suberin fatty acid (SFA) hydrolysate from birch outer bark estimated using GC-MS.

Compound	MEAN mg g^−1^ (o.d.)	SD
16:0 Fatty acid (palmitic acid)	0.83	0.06
Ferulic acid	5.48	1.42
17:0 fatty acid	0.12	0.02
18:2 Fatty acid (linoleic acid)	0.97	0.03
18:1 Fatty acid (oleic acid)	0.37	0.00
18:0 Fatty acid (stearic acid)	0.09	0.01
16-Hydroxy-16:0 acid	1.38	0.11
20:0 Fatty acid (arachidic acid)	0.19	0.03
1,16-Dioic-16:0 acid	3.36	0.25
18-Hydroxy-(9)18:1 acid	41.55	4.36
9,16- and 10,16-Dihydroxy-16:0 acids	13.70	0.22
18-Hydroxy-18:0 acid	1.31	0.17
1,18-Dioic-(9)18:1 acid	16.64	0.42
22:0 Fatty acid	0.00	0.00
1,18-Dioic-18:0 acid	6.93	0.53
9,18-Dihydroxy-(9)18:1 acid	5.22	0.02
9,10-Epoxy-18-hydroxy-18:0 acid	138.71	28.18
20-Hydroxy-20:1 acid	5.07	0.48
Dihydroxyoctadecanoic acid	3.73	0.13
20-Hydroxy-20:0 acid	8.60	1.52
1,20-Dioic-20:1 acid	2.89	0.04
24:0 Fatty acid	0.22	0.05
1,20-Dioic-20:0 acid	7.44	1.17
9,10,18-Trihydroxy-18:0 acid	130.53	40.72
(9)10-Chloro-10(9),18-dihydroxy-18:0 acid	22.99	11.72
22-Hydroxy-22:1 acid	1.21	0.08
22-Hydroxy-22:0 acid	11.29	3.17
1,22-Dioic-22:0 acid	13.59	7.10
24-Hydroxy-24:0 acid	0.00	0.00
Lupenone (lup-20(29)-en-3-one)	0.00	0.00
Sitosterol	0.66	0.08
Olean-12-ene-3,28-diol	0.94	0.18
Lupeol	0.24	0.04
Betulinol	20.49	2.22
Betulinic acid	18.17	2.49
Monogynol A (lupane-3b,20-diol)	0.00	0.00
Lupaen-3b,20,28-triol	0.00	0.00
Total identified (mg g^−1^)	484.91	25.07
Total eluted (mg g^−1^)	573.24	21.71

## Data Availability

Not applicable.

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
