# Peer review of "Suberin Fatty Acid Hydrolysates from Outer Birch Bark for Hydrophobic Coating on Aspen Wood Surface"

_polymers, 2022, doi:10.3390/polym14040832_

Round 1

Reviewer 1 Report

The review report, manuscript title: Suberin fatty acid hydrolysates from outer birch bark for hydrophobic coating on aspen wood surface. 

The manuscript is well written. There are some minor issues, that should be addressed.

Abbreviations: Please define during the first use, then use only abbreviations. FTIR and SEM-EDS should be also defined.

The figure before Introduction - I suggest moving it into the Materials and Methods section.

...large amounts of bark... please add some numbers for example annually in Europe?

The Introduction: Please add other ways of utilization of birch bark besides energy use.

I suggest merging two paragraphs in the Introduction section dealing with birch bark.

Please be more specific about wood samples preparation.

Please also statistical evaluation of results into Materials and Methods section.

The results part is well written.

In the Conclusions part, I suggest adding limitations of your research and implications for further research.

Author Response

Response to reviewer 1 comments 

The review report, manuscript title: Suberin fatty acid hydrolysates from outer birch bark for hydrophobic coating on aspen wood surface.

1. The manuscript is well written. There are some minor issues, that should be addressed.

We would like to thank the referee for reviewing our manuscript. We have revised the manuscript as per referee comments.

2. Abbreviations: Please define during the first use, then use only abbreviations. FTIR and SEM-EDS should be also defined.

Thank you for the suggestion, we have defined the abbreviations in very first appearance.

3. The figure before Introduction - I suggest moving it into the Materials and Methods section.

Thanks for the suggestion. We have moved figure (graphical abstract) to the materials and methods section and changed the subsequent figures number.

4....large amounts of bark... please add some numbers for example annually in Europe?

The global annual bark production is between 300 and 400 million cubic meters. Pásztory, Z., Mohácsiné, I. R., Gorbacheva, G., and Börcsök, Z. (2016). "The utilization of tree bark," BioRes. 11(3), 7859-7888.

5. The Introduction: Please add other ways of utilization of birch bark besides energy use.

Thanks for the suggestions, we have added more details about birch bark’s other utilization with new references.

6. I suggest merging two paragraphs in the Introduction section dealing with birch bark.

We have merged the two paragraphs dealing with birch bark.

7. Please be more specific about wood samples preparation.

We have added the details about wood samples preparations and dimensions in Materials section.

7. Please also statistical evaluation of results into Materials and Methods section.

Thank you for suggestions, we have added the details about statistical analysis in at materials and methodology part.

8. The results part is well written.

thank you for the compliment.

9. In the Conclusions part, I suggest adding limitations of your research and implications for further research.

We have added the following sentences in conclusions part:

The extraction process of suberin fatty acids from birch bark could impact the quality of bio-based polyesters. So, important considerations need to pay on the extraction process in future course of research work.T

Reviewer 2 Report

The paper presents an interesting experimental work on bark extracts, specifically suberin fatty acids hydrolysate (SFA) used as a wood coating with different cross-linking agents. The effect of two cross-linker on the coating properties was analysed. The topic is interesting and I recommend the publication after some revisions.

  1. The introduction should be updated with recent application of bark extracts: cite for example Polymers 2021, 13(24), 4380; https://doi.org/10.3390/polym13244380; Materials 2021, 14(24), 7774; https://doi.org/10.3390/ma14247774 and more others
  2. The quality of Figure 1 is scarce. The instrument screenshot should be replaced by a plot with well defined and labeled axes.
  3. The retention of the different coatings should be determined. Refers, for example, to the method reported in Materials and Design 30 (2009) 3303–3307
  4. The FTIR results reported in Figure 2 are not analysed for all the coatings. Please add more discussion.
  5. What about the scratch resistance and the thermal properties of the coating?
  6. In the authors opinion, are some of the proposed coatings suitable also for degraded wood?
  7. A discussion on the opportunity to prefer a cross-linker to another should be added to the conclusions

Author Response

Response to reviewer 2 comments 

1. The paper presents an interesting experimental work on bark extracts, specifically suberin fatty acids hydrolysate (SFA) used as a wood coating with different cross-linking agents. The effect of two cross-linker on the coating properties was analysed. The topic is interesting, and I recommend the publication after some revisions.

We would like the thank the referee for reviewing our manuscript and suggesting the further improvement.

2. The introduction should be updated with recent application of bark extracts: cite for example Polymers 2021, 13(24), 4380; https://doi.org/10.3390/polym13244380; Materials 2021, 14(24), 7774; https://doi.org/10.3390/ma14247774 and more others The quality of Figure 1 is scarce. The instrument screenshot should be replaced by a plot with well-defined and labeled axes.

We have added the suggested and some more reference in the introduction chapter. We have replaced the Figure 1 with new FTIR curve and now it is became Figure 2.

3. The retention of the different coatings should be determined. Refers, for example, to the method reported in Materials and Design 30 (2009) 3303–3307 The FTIR results reported in Figure 2 are not analysed for all the coatings. Please add more discussion.

We would like to thank the referee for important suggestion. At this stage of manuscript, it is hard to calculate the coating retention onto wood surface. However, add the more details in methodology chapter about the coatings weight depositions on wood surface. However, we have added more discussion in FTIR results mostly on SFA and combinations of cross-linking agents on wood surface.

4. What about the scratch resistance and the thermal properties of the coating?

We haven’t performed the scratch resistance and thermal properties of coating at this point of research. However, this is an important point, and we are in planning to perform TGA and scratch resistance of coatings for further research activities.

In the authors opinion, are some of the proposed coatings suitable also for degraded wood?

5. We are not sure, but we could consider this suggestion for test our coating formulations on degraded wood.

A discussion on the opportunity to prefer a cross-linker to another should be added to the conclusions

6. We have added the following sentences in the conclusion chapter:

In terms of crosslinker impact on SFA coating formulation, MA is mostly used in bio-based polyesters crosslinking and relatively successful in various application. OTS is very first time used as crosslinker for SFA and it perform better than MA. However, the combination of both MA and OTS performed better in compared to all combinations.

Round 2

Reviewer 2 Report

The authors addressed not all the comments by justifying the lack of facility or data at this point of review. However, I can accept the paper due to its significance